# Anti-Apoptotic and Anti-Inflammatory Effects of an Ethanolic Extract of *Lycium chinense* Root against Particulate Matter 10-Induced Cell Death and Inflammation in RBL-2H3 Basophil Cells and BALB/c Mice

**DOI:** 10.3390/plants11192485

**Published:** 2022-09-22

**Authors:** Jisun Lee, Jin Ree, Hyeon Jeong Kim, Hee Jin Kim, Woo Jung Kim, Tae Gyu Choi, Sanghyun Lee, Yun Ki Hong, Seong Bin Hong, Yong Il Park

**Affiliations:** 1Department of Biotechnology, Graduate School, The Catholic University of Korea, Bucheon 14662, Gyeonggi-do, Korea; 2Biocenter, Gyeonggido Business and Science Accelerator, Suwon 16229, Gyeonggi-do, Korea; 3Department of Biochemistry and Molecular Biology, School of Medicine, Kyung Hee University, Seoul 02447, Korea; 4Department of Plant Science and Technology, Chung-Ang University, Anseong 17546, Gyeonggi-do, Korea; 5Biomix Co., Ltd., Goyang-si 10442, Gyeonggi-do, Korea

**Keywords:** PM10, *Lycium chinense* root, basophils, apoptosis, inflammation

## Abstract

Particulate matters (PMs) from polluted air cause diverse pulmonary and cardiovascular diseases, including lung inflammation. While the fruits (Goji) of *Lycium* trees are commonly consumed as traditional medicine and functional food ingredients, the majority of their roots are discarded as by-products. To enhance the industrial applicability of *Lycium* roots, we prepared an ethanol extract (named GR30) of *L. chinense* Miller roots and evaluated its potential protective effects against particulate matter 10 (PM10)-induced inflammation and immune cell death. The GR30 treatment (0–500 μg/mL) significantly attenuated the PM10-induced cell cycle arrest, DNA fragmentation and mitochondria-dependent apoptosis in RBL-2H3 basophil cells. GR30 also significantly antagonized the PM10-induced expression of proinflammatory cytokines (IL-4, IL-13, and TNF-α) and COX2 expression through downregulation of MAPKs (ERK and JNK) signalling pathway. Oral administration of GR30 (200–400 mg/kg) to PM10 (20 mg/mL)-challenged mice significantly reduced the serum levels of IgE and the expression of TNF-α and Bax in lung tissues, which were elevated by PM10 exposure. These results revealed that the ethanolic extract (GR30) of *L. chinense* Miller roots exhibited anti-inflammatory and cyto-protective activity against PM10-induced inflammation and basophil cell death, and thus, it would be useful in functional food industries to ameliorate PM-mediated damage to respiratory and immune systems.

## 1. Introduction

Particulate matters (PMs) inhalation epidemiologically causes pulmonary and cardiovascular diseases, increasing PM-associated mortality and morbidity. In particular, exposure to PM influences the immune system, which increases diverse pathological risks, such as the development of cardiovascular diseases and inflammation and the reduction in lung function as well as diabetic symptoms [1,2]. In the lung, PM10 (particulate matter 10) was reported to elicit inflammatory processes and induces cell death [3,4]. Programmed cell death, especially apoptosis, is regarded as an important common factor for adverse clinical outcomes in the respiratory system [2,5]. Therefore, seeking effective materials that can mitigate PM-mediated inflammation and apoptotic death of respiratory and immune cells has been attracting topics of research on pulmonary and cardiovascular diseases and related medicinal and functional food industries.

Inflammation is a nonspecific immune response to any injury or attack and is normally a self-limiting process [6]. Generally, the first key sign of inflammation is the recruitment of various immune cells, such as neutrophils, macrophages and basophiles, from the vasculature into the injured tissue [2]. The PM was reported to induce severe lung inflammation in mice [5,7] and cytokine release, cytotoxicity and apoptosis in macrophages [8]. Although macrophages function as the first responders, basophils are pivotal responders for further defence mechanisms, such as allergic inflammation. Basophils play crucial roles in both immunoglobulin E (IgE)-dependent and IgE-independent allergic inflammation through their migration to the site of inflammation and secretion of various mediators, including cytokines, chemokines, and proteases. Basophils are known to produce large amounts of interleukin-4 (IL-4) and interleukin-13 (IL-13) in response to various stimuli, and IL-4 and IL-13 are essential in the induction of IgE synthesis [7,8]. Additionally, tumour necrosis factor-α (TNF-α), a key inflammatory mediator, induces other proinflammatory cytokines and endothelial adhesion molecules [9]. Thus, basophil cells are considered a good tool for studying the effect of environmental pollutants on chemical mediator release activity [10]. However, the pathophysiological relationship between PM-mediated inflammation and immune cell apoptosis and the role of basophils have not yet been extensively explored.

*Lycium chinense* (LC) and *Lycium barbarum* (LB) belong to the Solanaceae family and their fruits have long been widely used as various traditional herbal medicines and functional food ingredients under the common name Goji or Boxthorn in China, Gugija in Korea, Kuko in Japan and wolfberries in European countries. Various beneficial health effects of fruits (Goji berries) and leaves of these trees, such as hepatoprotective [11], antioxidant [12,13], immune enhancing [14], and hypoglycaemic and hypolipidaemic activities [13], have been reported. Regarding the roots of these traditional medicinal plants, only a small portion of their bark (Lycii Radicis Cortex, LRC) has been traditionally utilized in Korea (under the name of Jigolpi) and China (known as Digupi) for the cure of various symptoms. However, other than the use of their bark (LRC), their whole roots have attracted relatively less attention and have just been discarded as by-products from the Goji farm yards, although they still contain various bioactive substances [15,16,17,18]. Several recent studies reported that the roots of *Lycium* species also exhibit various beneficial health effects [15], such as antioxidant [19], antitumour [20], anti-osteoclastic [21], antimicrobial [22], and immunomodulatory activities [23]. Thus, to increase industrial utility values of *Lycium* roots, such as health-beneficial functional foods or dietary supplements, it would be worth searching for their various biological and medicinal potentials.

Recently, accumulating evidence showed that the roots of *Lycium* species also contain various anti-inflammatory metabolites such as kukoamine A [18], kukoamine B [16], and lyciumin [17]. The PMs are now known to induce severe lung inflammation [5,7] and cytotoxicity in many cell types including immune cells [4]. Thus, we hypothesized that the roots of these medicinal plants may also have potential protective roles against PM-mediated inflammatory complications and immune cell death. Until now, no specific studies on the protective effects of *Lycium* roots through anti-inflammatory and cytoprotective activities against PM-induced inflammation and immune cell death, have been reported. In this context, in the present study, we prepared a 30% ethanol extract (named GR30 in this study) of the whole roots of *L. chinense* Miller and evaluated its anti-inflammatory and anti-apoptotic properties against PM10-induced inflammation and basophil immune cell death using basophilic leukaemia RBL-2H3 cells and PM10-challenged mice as an in vivo animal model system.

## 2. Results

### 2.1. Effects of PM10 and GR30 on RBL-2H3 Cell Viability

Whether PM10 and GR30 are detrimental to RBL-2H3 cell growth was assessed by measuring the cell viability with the trypan blue exclusion assay. Cells were treated with PM10 (100 μg/mL) or GR30 (25–500 μg/mL) for 48 h, harvested through trypsinization, stained with trypan blue, and the viable cells were counted using a haemocytometer. As shown in Figure 1A, the viability of RBL-2H3 cells was not affected by GR30, suggesting that GR30 (up to 500 μg/mL) was not cytotoxic to the basophils. However, PM10 treatment significantly decreased the cell viability to 63.9% (36.1% reduction) compared to the untreated control (Con, 100%) (Figure 1B). To investigate the preventive effect of GR30 against PM10-induced cytotoxicity, cells were pretreated with varying concentrations of GR30 (25–500 μg/mL) for 2 h and then incubated with PM10 (100 μg/mL) for 48 h. The decreased cell viability by PM10 (63.9%) was significantly and dose-dependently recovered by GR30, up to 104.7% at 500 μg/mL. These results suggested that PM10 exerts cytotoxicity to basophil cells, but GR30 has preventive effects against PM10-induced basophil cell death.

### 2.2. Effects of PM10 and GR30 on Cell Cycle Arrest and DNA Fragmentation

Cells were incubated with GR30 for 2 h, then treated with PM10 for 48 h, and then the expression levels of p21 (a tumour suppressor acting as a blocker of cell cycle progression) and CyclinD1 (an activator of cell cycle progression) were determined by Western blot analysis (Figure 2A). As shown in Figure 2B, the CyclinD1 expression level was significantly decreased by PM10 treatment (0.58 fold) compared with untreated control cells (Con, 1 fold), whereas the PM10-caused decrease in CyclinD1 expression was dose-dependently reversed by GR30 pretreatment. While the p21 expression level significantly increased in the PM10-only-treated cells (1.67 fold), GR30 significantly and dose-dependently downregulated its expression to 0.47 fold at 500 μg/mL (Figure 2C). These results suggested that GR30 treatment untied the PM10-induced cell cycle arrest through downregulation of p21 expression and upregulation of CyclinD1 expression.

The protective effect of GR30 against the PM10-induced cytotoxicity on the basophil cells was further confirmed by analysing the level of DNA fragmentation, which is a hallmark of apoptosis (Figure 2D). Nuclei in RBL-2H3 cells were simultaneously monitored by PI staining. Upon treatment with PM10 only, the number of TUNEL-positive cells (shown as green dots) significantly increased compared with the control normal cells (Con), indicating that PM10 causes apoptotic cell death of basophils by induction of DNA fragmentation. Pretreatment with GR30 dose-dependently decreased the number of TUNEL-positive cells, showing that GR30 effectively inhibited the PM10-induced DNA breaks in basophil cells (Figure 2D).

### 2.3. GR30 Antagonizes PM10-Induced Apoptotic Cell Death via Mitochondria-Dependent Apoptosis in RBL-2H3 Cells

To determine whether PM10 induces mitochondria-dependent apoptosis, cells were incubated with GR30 (25–500 μg/mL) for 2 h and then treated with PM10 for 48 h and Western blot analysis was performed (Figure 3A). The PM10 treatment significantly increased the expression of Bax up to 1.9 fold (Figure 3B) and cleaved caspase-3 up to 1.48 fold (Figure 3E), while that of Bcl-2 decreased up to 0.74 fold (Figure 3C), compared to the untreated control (Con, 1 fold). However, pretreatment with GR30 markedly reversed the PM10-induced elevated levels of Bax expression to 1.02 fold (Figure 3B) and cleaved caspase-3 to 0.7 fold (Figure 3E), compared to the PM10-untreated normal control cells (Con, 1 fold), at 500 μg/mL. Moreover, GR30 restored the Bcl-2 expression level, which was downregulated by PM10 treatment (Figure 3C). PM10 treatment remarkably increased the ratio up to 2.45 fold compared to untreated normal cells (Con, 1 fold), but GR30 significantly and dose-dependently reduced the PM10-induced upregulated level of this ratio (Figure 3D). These results clearly demonstrated that PM10 induces apoptotic death of basophil cells through activation of mitochondria-dependent apoptosis pathway in RBL-2H3 cells and that GR30 protects basophil cells from the toxic effects of PM10 by attenuating the mitochondria-dependent apoptotic pathway activation triggered by PM10 exposure.

### 2.4. GR30 Suppresses PM10-Induced Inflammatory Cytokine and COX2 Secretion in Basophils

Whether PM10 and GR30 could modulate the mRNA expression of proinflammatory cytokines such as IL-4, IL-13, and TNF-α in RBL-2H3 cells were examined (Figure 4A). The cells were treated with GR30 (25–500 μg/mL) for 2 h and then incubated with PM10 (100 μg/mL) for 48 h. The IL-4 (Figure 4B), IL-13 (Figure 4C) and TNF-α (Figure 4D) mRNA levels in the PM10 group (treated with PM10 only) were markedly increased up to 5.19, 1.96 and 1.41 fold, respectively, compared to the PM10-untreated control cells (Con, 1 fold). However, GR30 treatment significantly suppressed the expression of IL-4, IL-13 and TNF-α cytokines in the PM10-treated cells, up to 3.27 (Figure 4B), 1.32 (Figure 4C) and 0.88 fold (Figure 4D), respectively, at 500 μg/mL, compared to the PM10-untreated control group (Con, 1 fold). Furthermore, as shown in Figure 4E, while the PM10 treatment also significantly increased the expression of COX2 mRNA by1.27 fold, GR30 pretreatment dose-dependently reduced its expression, by 0.67 fold at 500 μg/mL, compared to the PM10-untreated control group (Con, 1 fold). These results suggested that PM10 induces inflammation through stimulation of basophils and that GR30 exerts anti-inflammatory effects under the PM10-caused inflammatory condition.

### 2.5. GR30 Downregulates the PM10-Induced MAPK Pathway Activation

To determine whether PM10 and GR30 could alter the MAPKs pathway activation, their effects on the phosphorylation levels of MAPKs (ERK and JNK) were examined (Figure 5). The cells were incubated with GR30 (25–500 μg/mL) for 2 h and treated with PM10 for 48 h, and the phosphorylation levels of ERK and JNK were determined by Western blot analysis (Figure 5A). While PM10 treatment did not affect the total protein levels of ERK and JNK, it markedly enhanced the phosphorylation levels of ERK (Figure 5B) and JNK (Figure 5C) up to approximately 2.13 and 2.14 fold, respectively, compared to the PM10-untreated control group (Con, 1 fold). However, GR30 pretreatment prior to PM10 exposure, the phosphorylation of ERK and JNK was inhibited up to 0.69 (Figure 5B) and 1.18 fold (Figure 5C) at 500 μg/mL, respectively, compared to PM10-only-treated cells. These results implied that PM10 induces inflammation through activation of MAPK pathway and that GR30 exerts anti-inflammatory effects against PM10-induced inflammation through suppression of MAPK pathway activation.

### 2.6. GR30 Suppresses PM10-Induced Serum IgE Production and TNF-α and Bax Expression in Lung Tissues

On day 22, starting from the first day of sample administration (Figure 6A), all sample-treated groups, PM10 only (20 mg/mL, 30 min exposure), PM10 + GR30 (200 mg/kg), and PM10 + GR30 (400 mg/kg), did not change the body weight of the mice compared to the untreated control group (data not shown). Exposure to PM10 resulted in an increase in the serum IgE level compared with that found in the untreated normal control group (Con) (Figure 6B). This increased serum IgE level in the PM10-only-treated group significantly and dose-dependently decreased in the GR30-treated groups (GR30-L, 200 mg/kg; GR30-H, 400 mg/kg). In addition, the mRNA level of TNF-α significantly increased by 2.86 fold in the lung tissues of PM10-challenged mice compared with the untreated control group (Con, 1 fold) (Figure 6C). However, this elevated TNF-α expression level was significantly reversed in the GR30 administered mice up to 1.38 fold at 400 mg/kg (*p* < 0.001) compared with those observed in the PM10-only-challenged mice (PM10 group) (Figure 6C). Meanwhile, the expression levels of Bax (a proapoptotic factor) in lung tissue markedly decreased in GR30-treated mice group by 2.05 fold at 400 mg/kg (*p* < 0.01) compared with the PM10-only-challenged group (4.69 fold) (Figure 6D), which was consistent with the in vitro results with RBL-2H3 basophil cells as observed in Figure 3. Taken together, these results demonstrated that PM10 causes apoptotic death of basophil cells and inflammation and that GR30 effectively attenuates the PM10-induced inflammatory basophil cell death in vitro and in vivo.

### 2.7. Chemical Profile of GR30 Determined by HPLC and LC-MS Analysis

Betaine is present in the fruits of *L. chinense* Miller and used as an index compound for various *Lycium*-based herbal medicines and food products [24,25]. As shown in Figure 7, HPLC analysis confirmed the presence of betaine in GR30, which is a 30% ethanol extract of the roots of *L. chinense* Miller, containing 5.2 mg per g of GR30. The chemical profile of major compounds present in GR30 was determined by UHPLC-MS analysis. The LC-MS/MS analysis identified and confirmed several important anti-inflammatory metabolites in GR30. As shown in Figure 8A, chromatographic separation of compounds showed various peaks, demonstrating the presence of different chemotypes in GR30. Of them, five major compounds shown in Figure 8A, eluted at RT of 5.14 min (P1), 8.37 (P2), 9.61 (P3), 9.98 (P4), and 11.06 (P5), were identified as kukoamine A or kukoamine B (P1), 4-[4-(tert-butoxycarbonyl)piperazin-1-yl]benzoic acid (P2), *n*-caffeoyltyramine (P3), lyciumin A (P4), and coumarin 314 (P5), respectively (Figure 8B and Table 1). These compounds, including betaine, have been reported for their anti-inflammatory activities [18,25,26], and thus, the observed anti-inflammatory and protective effects of GR30 against PM10-mediated inflammation and basophil cell death could be attributed to these compounds present in GR30.

Chemical profile of major compounds present in GR30 was determined by UHPLC-MS analysis. Samples dissolved in 80% methanol were fractionated on a C_18_ column by UHPLC and major compounds were resolved into five major peaks with RT as follows: 5.14 min (P1), 8.37 (P2), 9.61 (P3), 9.98 (P4), and 11.06 (P5). The MS analysis was performed with polarity switching, with following parameters for MS/MS scan: *m*/*z* range of 150–1500; collision-induced dissociation energy of 45%; data-dependent scan mode.

## 3. Discussion

Seeking effective therapeutic and functional materials that can mitigate inflammation and cell death caused by particulate matters (PMs) originated from the polluted environment has been increasingly important topics of research on human health and bioindustries. The results of this study demonstrated that PM10 treatment significantly evokes apoptotic death of basophil immune cells and inflammation in vitro and in vivo and that the GR30 effectively ameliorates the PM10-mediated apoptotic death of basophils and inflammatory complications through its anti-inflammatory and anti-apoptotic activities.

PM10 treatment significantly reduced the viability of RBL-2H3 basophil cells and this cytotoxicity of PM10 was mediated by cell cycle arrest through downregulation of CyclinD1 expression and upregulation of p21 expression and that GR30 treatment untied the PM10-induced cell cycle arrest through downregulation of p21 expression and upregulation of CyclinD1 expression. The ratio of Bax to Bcl-2 protein determines cell survival or death after apoptotic stimuli [27]. PM10 increased the ratio of Bax/Bcl-2 and cleaved caspase-3 expression, which were noticeably attenuated by treatment with GR30, indicating that GR30 suppresses the PM10-induced mitochondria-dependent apoptosis, thereby rescuing basophil cells from PM-mediated cytotoxicity. Similarly to our present results from GR30, the *L. chinense* root extract, it was also reported that the fruit extracts of *L. barbarum* attenuate alcoholic cellular injury of hepatic cells through the reduction in cellular apoptosis [28].

The PM was reported to induce severe lung inflammation [5,7] and to induce cytokine release, cytotoxicity and apoptosis in macrophages [8]. Our findings on the cyto-protective property of GR30 raised one question, namely, how GR30 could mitigate the PM10-induced apoptotic cell death of basophils. As mentioned earlier, it is well known that PM10 induces both apoptosis and inflammation, but the relationship between these two biological processes has not been clearly addressed. The production of cytokines is a key event in the initiation and regulation of immune responses [12]. However, overproduction of proinflammatory cytokines could result in acute adverse health effects, including pathological lung injury and inflammatory diseases [5]. Basophils are known to release proinflammatory cytokines such as IL-4, IL-13 and TNF-α in response to antigens [29,30]. Our present study revealed that GR30 suppressed the production of inflammatory cytokines (TNF-α, IL-4 and IL-13) induced by PM10 in RBL-2H3 basophil cells. Furthermore, GR30 markedly suppressed the phosphorylation of MAPKs under PM10 exposure. The release of proinflammatory cytokines is affected by the activation of MAPKs in RBL-2H3 cells [31]. Therefore, our findings indicate that GR30 significantly inhibits the PM10-induced inflammatory response in basophils through its direct involvement in the reduction in proinflammatory mediator production. Previously, Diao et al. reported that activated MAPKs participate in the PM10-induced cytokine expression in mammalian cells [32]. Additionally, our present study also showed that PM10 treatment significantly increased the mRNA expression of COX2 in RBL-2H3 cells, but GR30 pretreatment inhibited the PM10-induced COX2 expression. COX is the enzyme that converts arachidonic acid to prostaglandins (PGs), which are a group of physiologically active lipid compounds called eicosanoids and are involved in inflammation. Of the two isoforms of COX, COX1 is required for normal physiologic functions such as gastrointestinal cytoprotection and platelet activity, whereas COX2 is mainly an inducible enzyme and is involved primarily in the regulation of inflammation. Thus, the inhibition of COX2 enzyme expression can effectively ameliorate pathophysiologic inflammation [33]. It has been reported that PM10 stimulated the expression of proinflammatory cytokines and COX2 enzyme in A549 lung epithelial cells, and an extract of a plant *Rosa laevigata* could suppress their expression [34]. In our present study, the results with COX2 were in consistent with the results shown with the inhibition of PM10-induced expression of cytokines (IL-4, IL-13, and TNF-α), providing additional evidence that PM10 causes inflammation in RBL-2H2 basophil cells and GR30 exhibits anti-inflammatory effects in RBL-2H3 cells by antagonizing the PM10-induced proinflammatory cytokines expression and COX2 enzyme expression.

PM10 is a common inflammation inducer from the airways to the lung that elevates IgE and inflammatory cytokine levels [3,29]. In the present study, the observed in vitro cytotoxicity and stimulation of inflammatory response on the RBL-2H3 basophil cells by PM10 was equally observed in PM10-exposed mice and the in vitro anti-inflammatory and cytoprotective activity of GR30 was also shown to be equally effective in vivo. IgE promotes inflammation progression by binding to and activating FcεR, a high-affinity receptor for Fc region of IgE, in macrophages and mast cells [35]. Basophils infiltrate peripheral tissues during allergic inflammation and express FcεR1, similar to mast cells. Thus, interfering with the binding between IgE and FcεRI is considered a straightforward strategy for inhibiting the allergic and asthmatic reactions [36,37], which is a good target for the development of anti-inflammatory, anti-allergic functional foods and therapeutics. It was suggested that 5-hydroxymethylfurfural (5-HMF), an active component of *L. chinense* fruits, was useful for treating and preventing allergic diseases by blocking the cross-linking of IgE and IgE binding to the FcεRI receptor [10]. Interestingly, our results showed that the serum IgE levels of mice exposed to PM10 increased, but this increase was noticeably reversed by the GR30 treatment. Thus, we suggest that GR30 directly blocks IgE secretion into the bloodstream as an inhibitory mechanism for PM10-induced allergic inflammation.

Meanwhile, it was reported that in IgE-mediated inflammatory responses, TNF-α is released to serum in sufficient amounts to provoke biological responses such as granulocyte chemotaxis and upregulation of TNF-α mRNA in eosinophils and other cells [38]. These findings suggest that TNF-α is an important modulator of the pathogenesis of allergic respiratory reactions. Consistently, our results also showed that the pattern of the TNF-α mRNA expression level was similar to the serum IgE level in the blood of the PM10-exposed mice. Accordingly, the increased expression of Bax in the lung tissue of PM10-inhaled mice was restored by GR30 administration, indicating that GR30 ameliorates apoptotic damage in the lung tissues of mice exposed to PM10. It was also reported that PM10-induced inflammation could stimulate immune cell death in the lung, possibly leading to severe pathological lung dysfunction [5,39]. Therefore, the results of our present study clearly indicated that GR30 ameliorates PM10-induced inflammation and apoptotic death of basophils, thereby possibly protecting lung tissues from PM10-induced damage. The observed anti-inflammatory effects of GR30 may be achieved by multiple mechanisms, including a reduction in IgE secretion and the expression of TNF-α and Bax.

HPLC and LC-MS analysis confirmed that the GR30 preparation contains various anti-inflammatory compounds including betaine, kukoamine A or kukoamine B, 4-[4-(tert-butoxycarbonyl)piperazin-1-yl]benzoic acid, *n*-caffeoyltyramine, lyciumin A, and coumarin 314. Betaine is a famous metabolite for its various bioactivities including modulation of oxidative stress, inflammation, apoptosis, and autophagy [25]. Betaine is present in the fruits of *L. chinense* Miller and is currently used as an index compound for various *Lycium*-based herbal medicines and food products [24,25]. In addition to fruits, the presence of betaine in the root bark (Lycii Radicis Cortex, LRC) of *L. chinense*, which has been commonly consumed as a herbal folk medicine called Jigolpi in Korea or Digupi in China, was also reported [40]. Recently, its presence in a water extract of the whole root of *L. chinense* Miller was also confirmed [23]. Additionally, the results of LC-MS analysis of GR30 revealed the presence of various chemotypes including 5 major compounds, kukoamine A or kukoamine B, 4-[4-(tert-butoxycarbonyl)piperazin-1-yl]benzoic acid, *n*-caffeoyltyramine, lyciumin A, and coumarin 314. It was reported that the *Lycium* roots contain anti-inflammatory metabolites such as kukoamine A [18], kukoamine B [16], and lyciumin [17]. Coumarins are a group of natural secondary metabolites found in a wide range of plants and have attracted great attention for their various bioactivities including anti-inflammatory activity [26]. *n*-Caffeoyltyramine is also found in the *Lycium* roots and exhibits anti-inflammatory activity [41]. Therefore, the observed anti-inflammatory and protective effects of GR30 in the present study against PM10-induced inflammation and apoptotic cell death of basophils might be, at least partly, attributed to these compounds present in GR30 preparation. However, to confirm this hypothesis, further research on, for example, isolation of individual compounds and characterization of their anti-inflammatory and cyto-protective activities against PM10-induced inflammatory apoptotic death of basophil cells, needs to be performed in the future. In addition to these compounds identified in the present study, further study on mining additional compounds in the GR30, if any exist, that have similar anti-inflammatory and cytoprotective activities against the toxic effect of PM10 would augment the applicability of GR30 as an active ingredient for health foods as well.

In summary, our findings in the present study revealed for the first time that PM10 causes inflammatory apoptotic death of basophil immune cells and that the GR30, an ethanolic extract of whole roots of *L. chinense*, effectively ameliorates the PM10-induced inflammation and apoptotic cell death of basophils. GR30 could rescue PM10-stimulated RBL-2H3 cells from cell cycle arrest, DNA fragmentation, mitochondria-dependent apoptosis and inflammation by inhibiting the phosphorylation of MAPKs (ERK and JNK) and production of proinflammatory cytokines such as TNF-α, IL-4, and IL-13 and COX2 enzyme expression. GR30 also reduced the serum IgE production and suppressed the expression of proinflammatory cytokine, TNF-α, and apoptotic regulator, Bax, in the lung tissues of PM10-challenged mice. GR30 contained several anti-inflammatory metabolites, such as betaine, kukoamine A or kukoamine B, 4-[4-(tert-butoxycarbonyl)piperazin-1-yl]benzoic acid, *n*-caffeoyltyramine, lyciumin A, and coumarin 314. The presence of these compounds in GR30 may suggest their applicability as indicator compounds of any GR30-based functional food products or therapeutic herbal preparations in the future. Finally, our findings in the present study may provide a better insight on the understanding of pathophysiological mechanisms underlying PM10-induced adverse effects on immune cells, especially the basophils, and lung inflammation. From this finding, we suggest that the GR30 could be used as a potent functional ingredient in health foods or as a therapeutic agent to prevent and/or ameliorate particulate matter (PM)-induced damage to our immune system and inflammatory lung diseases.

## 4. Methods and Materials

### 4.1. Materials

Minimum essential medium (MEM) and foetal bovine serum (FBS) were purchased from Welgene (Seoul, Korea). Penicillin-streptomycin was obtained from Gibco (Grand Island, NY, USA). Trypan blue, PM10 (particulate matter 10, ≤ 10 µm in diameter) and betaine were obtained from Sigma Aldrich (St. Louis, MO, USA). Mouse IgE ELISA kit was obtained from R&D Systems (Minneapolis, MN, USA). Total RNA extraction and PCR premix kits were obtained from iNtRON Biotechnology (Seongnam, Gyeonggi-do, Korea). The cDNA reverse transcription kit was purchased from Applied Biosystems (Foster City, CA, USA). Antibodies against cleaved caspase-3, phospho-ERK (*p*-ERK), phospho-JNK (*p*-JNK), total ERK (t-ERK), total JNK (t-JNK), and β-actin were obtained from Cell Signaling Technology (Danvers, MA, USA). Bax, CyclinD1, and p21 antibodies were acquired from Santa Cruz Biotechnology (Santa Cruz, CA, USA).

### 4.2. Preparation and Major Chemical Profile of GR30

The roots of *L. chinense* were obtained from Cheongyang-gun, Chungcheongnam-do, Korea. The ethanol extract (GR30) of *L. chinense* roots was prepared and supplied by Biomix Inc. (Goyang, Korea). Briefly, the whole root parts of *L. chinense* tree were collected and washed with excess tap water and air-dried. The dried roots were crushed into powder and decocted twice with 30% ethanol on repeated reflux extraction. The extract was prepared in powder form using a lyophilizer (Ilshin Bio Base, Dongducheon, Korea), tentatively named GR30 (30% ethanol extract of *L. chinense* roots), and stored at 4 °C until use. Chemical profile analysis of GR30 was performed by HPLC and LC-MS.

The presence of betaine in GR30 was analysed using HPLC system (1260 series, Agilent, Santa Clara, CA, USA). Samples were dissolved in dH_2_O and fractionated on a Repromer ES column (250 × 4.6 mm, Dr. Maisch, Munich, Germany) and eluted at 0.7 mL/min in a gradient mode with mobile phase composed of water and acetonitrile. Other major compounds were identified by high resolution liquid chromatography-mass spectrometry (LC-MS). GR30 (100 mg) was mixed with 80% methanol and filtered using 0.22-μm Spin-X centrifuge tube filter (Sigma, Saint Louis, MO, USA) at 13,000 rpm for 3 min. After removal of methanol by evaporation, the filtrate was lyophilized and stored at −20 °C. The powdered sample was dissolved in methanol and analysed using an ultra-high-performance liquid chromatography (UHPLC) system (LTQ Orbitrap XL; Thermo Electron Co., Waltham, MA, USA) equipped with ACQUITY UPLC^®^ BEH C_18_ column (2.1 × 150 mm, 1.7 μm) at 40 °C with mobile phases A (water with 0.1% formic acid) and B (acetonitrile with 0.1% formic acid). The MS analysis was performed with polarity switching, with following parameters for MS/MS scan: *m*/*z* range of 150–1500; collision-induced dissociation energy of 45%; data-dependent scan mode. High resolution mass spectra were analysed with XCALIBUR software (Thermo Fisher Scientific, Waltham, MA, USA).

### 4.3. Cell Culture and Cytotoxicity Assay

Rat basophilic leukaemia RBL-2H3 cells obtained from the Korean Cell Line Bank (Seoul, Korea) were grown in minimum essential medium (MEM) supplemented with 1% penicillin-streptomycin and 10% FBS at 37 °C in a humidified incubator (5% CO_2_, 95% air). The cells (1 × 10^5^ cells/well) plated on 12-well microplates were pretreated with GR30 (25~500 μg/mL) for 2 h and then incubated with PM10 for 48 h. Cell viability was measured using the trypan blue exclusion method [42]. After the growth, both adherent and suspended cells were collected, mixed with 0.4% trypan blue buffer, transferred to the haemocytometer and then observed under a microscope (CK40-F200; Olympus, Kyoto, Japan). Blue-stained cells were counted as dead cells. The viable cells were unstained, while dead cells were stained blue.
(1)$$\%\;\mathrm{Cell}\;\mathrm{viability}=\left(1\;-\;\frac{\mathrm{Number}\;\mathrm{of}\;\mathrm{dead}\;\mathrm{cells}}{\mathrm{Number}\;\mathrm{of}\;\mathrm{total}\;\mathrm{cells}}\right)\times \;100$$

### 4.4. Animals, Diets and Experimental Protocol

Male BALB/c mice (5 weeks old) were purchased from Orientbio (Gyeonggi-do, Korea) and housed in an air-conditioned (21–25 °C) and humidity-controlled room with a 12 h/12 h on–off light. Animal care and handling were conducted under protocols approved by the Committee on Animal Experimentation of the Catholic University of Korea (Approval number 2017-004-02). A schematic diagram of the treatment schedule is presented in Figure 6A. The mice were randomly divided into four groups (8 mice/group) and fed normal diets (AIN-93G, 15.8% calories from fat). Mice were orally administered PBS (control group, PM10 group) or GR30 (200–400 mg/kg/day in 200 μL PBS for 21 days daily. The PM10-challenged mice were exposed to 20 mg/mL (*w*/*v*, in PBS) for 30 min by inhalation using a compressor nebulizer (0.4 mL/min, NE-C28, Omrom, Tokyo, Japan) and neglected for 2 h. On day 22, starting from the first day of sample administration, the mice were euthanized by IP injection of a 3:2 mixture of Tiletamine-Zolazepam (Zolletil50, Virbac S. A, Carros, France) and Xylazine (Rompun, Bayer, Leverkusen, Germany). The blood samples were collected and stored at −80 °C until the cytokine assays were performed. The lungs were rapidly removed, frozen in liquid nitrogen, and stored at −80 °C for total RNA and protein extraction.

### 4.5. Total RNA Extraction and Reverse Transcription PCR (RT-PCR)

Total RNA was extracted using TRIzol reagent (Invitrogen, Carlsbad CA, USA) and reverse-transcribed using a Power cDNA synthesis kit (iNtRON Biotech, Seongnam, Korea) with an oligo (dT)15 primer. The primers used are listed in Table 2. RT-PCR was performed with a Maxime PCR PreMix kit (iNtRON Biotech). PCR products were separated on a 1.5% agarose gel, stained with ethidium bromide and visualized using a UV transilluminator. The mRNA expression level of cytokines (IL-4, IL-13 and TNF-α) and cyclooxygenase 2 (COX2) was represented as the ratio of the densitometric measurement of the indicated mRNA to the housekeeping control gene β-actin.

### 4.6. Western Blot Analysis

Cells were precultured in 6-well culture plates, treated with GR30 for the indicated times, and washed with PBS. Cells were scraped in RIPA buffer (50 mM Tris, 150 mM NaCl, 2 mM EDTA, 1% Triton X-100, 0.1% SDS, pH 7.8) containing phosphatase inhibitor and protease inhibitor cocktail (Sigma, St. Louis, USA). Cell lysates were centrifuged at 13,000 rpm for 15 min, at 4 °C, to obtain intracellular proteins. The lung tissues were homogenized at 4 °C in RIPA buffer containing phosphatase inhibitors and protease inhibitors. Protein was measured by Bradford assay. Proteins (15–30 μg) were separated on a 10% or 12% SDS–PAGE gel and transferred to nitrocellulose membranes, blocked with 5% skim milk for 1 h, and incubated with primary antibodies for 2 h, at room temperature. Membranes were then incubated with horseradish peroxidase (HRP)-conjugated secondary antibodies, at room temperature, for 2 h and visualized using enhanced chemiluminescence (Abclon, Seoul, Korea). The relative amount of proteins was quantified by ImageJ software from the NIH (Bethesda, MD, USA).

### 4.7. TUNEL Assay

Apoptotic cell death was monitored using the TUNEL assay kit (Biovision, Mountain View, CA, USA). Briefly, RBL-2H3 cells were pretreated with GR30 (100, 250 and 500 μg/mL) for 2 h and then incubated with PM10 for 48 h. The cells were fixed with 1% paraformaldehyde for 15 min and incubated in 50 μL TUNEL enzyme and TUNEL label mixture for 1 h, at 37 °C, in a humidified atmosphere in the dark. The cells were visualized using a Zeiss LSM800 confocal laser scanning microscope (Carl Zeiss, Germany, X400).

### 4.8. Measurement of IgE Production

The serum IgE concentrations were determined using a mouse IgE ELISA kit (eBioscience, San Diego, CA, USA) at 450 nm. The difference between the sample absorbance and the mean of the negative control absorbance was expressed as the result.

### 4.9. Statistical Analysis

Each experiment was performed at least in triplicate, and data from independent experiments are expressed as the means ± SD. Comparisons between two groups were performed with unpaired Student’s *t*-test. Nonparametric Wilcoxon tests were performed to examine the changes between the GR30 treated group / Con or PM10. Multiple comparisons between experimental groups were determined by one-way ANOVA followed by Duncan’s multiple range test or Dunnett’s multiple comparison test. All statistical analyses were performed using SAS 9.4 Statistical Analysis Systems software package (SAS Institute Inc., Cary, NC, USA), and statistical significance was considered when *p* < 0.05.

## Figures and Tables

**Figure 1 plants-11-02485-f001:**
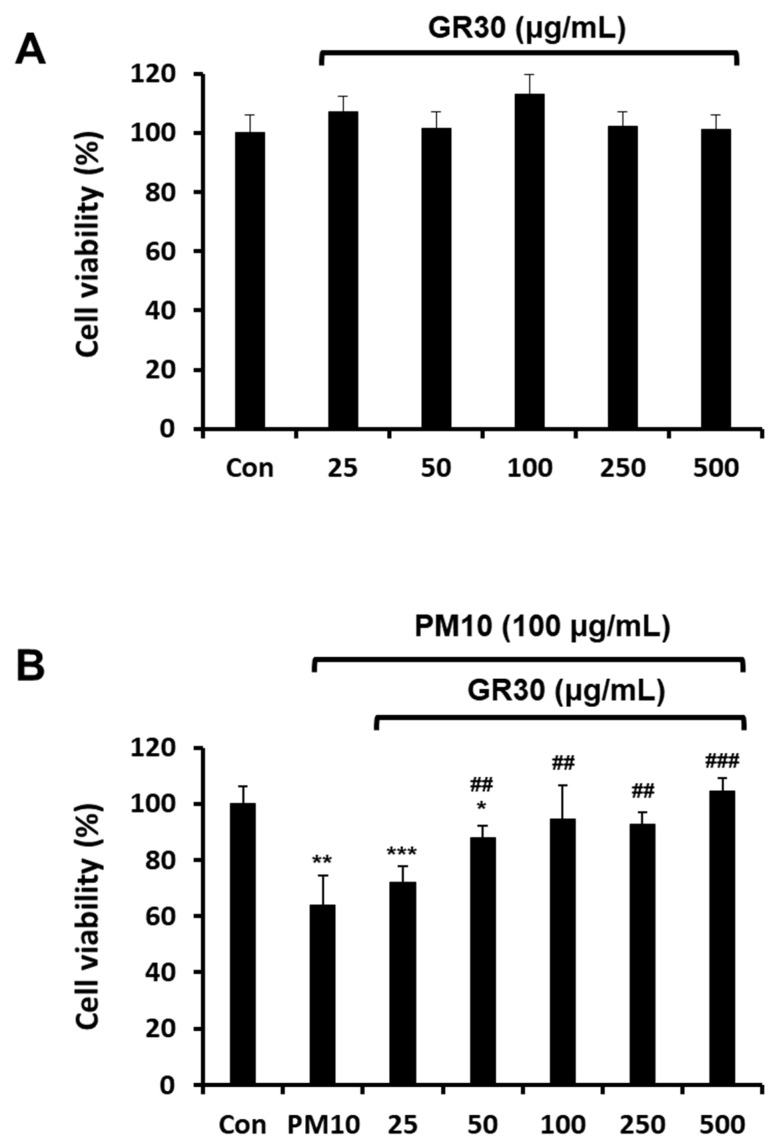
Effects of PM10 and GR30 on the viability of RBL-2H3 cells. (**A**) Cells were treated with GR30 (0–500 μg/mL) for 48 h. Cell viability was measured by the trypan blue exclusion method. (**B**) The cells were pretreated with GR30 (25~500 μg/mL) for 2 h, incubated with PM10 (100 μg/mL) for 48 h and then the cell viability was measured. Each value was presented as the means ± SD (*n* = 3). *, *p* < 0.05; **, *p* < 0.01; ***, *p* < 0.001, compared to the control cells (Con). ##, *p* < 0.01; ###, *p* < 0.001, compared to the PM10-treated cells (PM10).

**Figure 2 plants-11-02485-f002:**
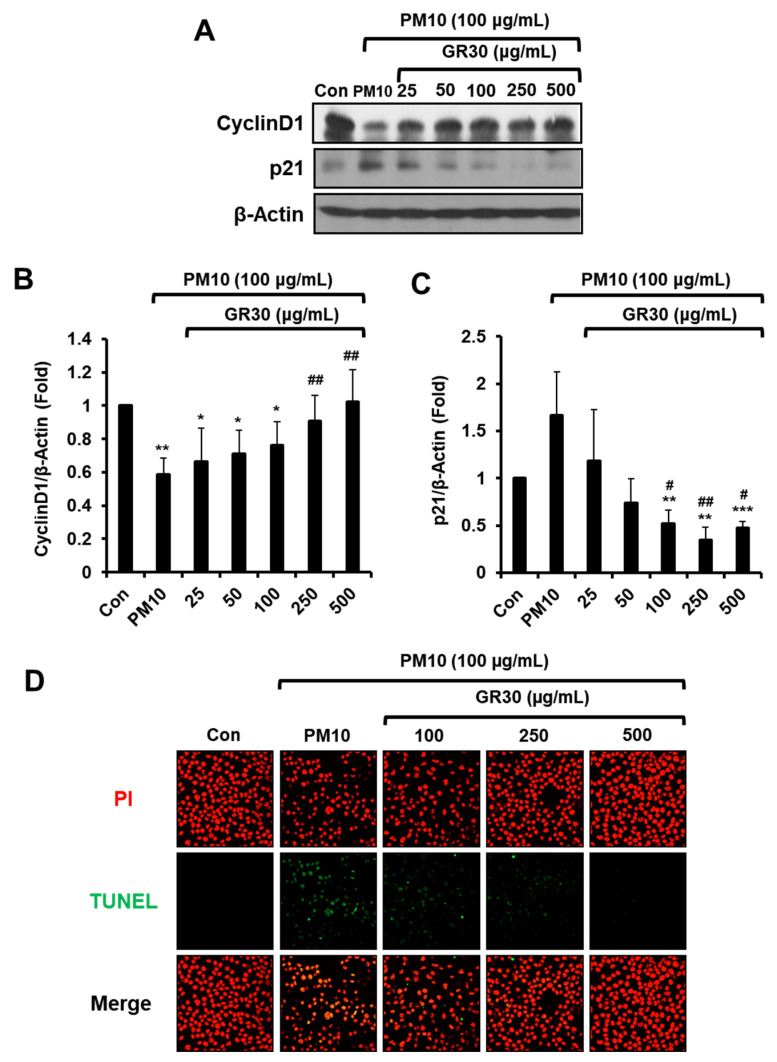
Effects of PM10 and GR30 on cell cycle arrest and DNA fragmentation. (**A**) Cells were pretreated with GR30 (25–500 μg/mL) for 2 h and then incubated with PM10 for 48 h. Cell lysates were analysed by Western blot analysis for p21 and cyclinD1. β-Actin was used as a loading control. (**B**,**C**) Protein levels of (**B**) cyclinD1 and (**C**) p21 were quantified by densitometric analysis. The data are expressed as the fold normalized to the untreated control (Con). Each value was presented as the means ± SD (*n* = 3). *, *p* < 0.05; **, *p* < 0.01; ***, *p* < 0.001, compared to the untreated control cells (Con). #, *p* < 0.05; ##, *p* < 0.01, compared to the PM10-treated cells (PM10). (**D**) DNA fragmentation was assessed by TUNEL staining (green) and counterstaining with PI (red). The stained cells were mounted with a mounting solution and visualized using fluorescence microscopy (X400).

**Figure 3 plants-11-02485-f003:**
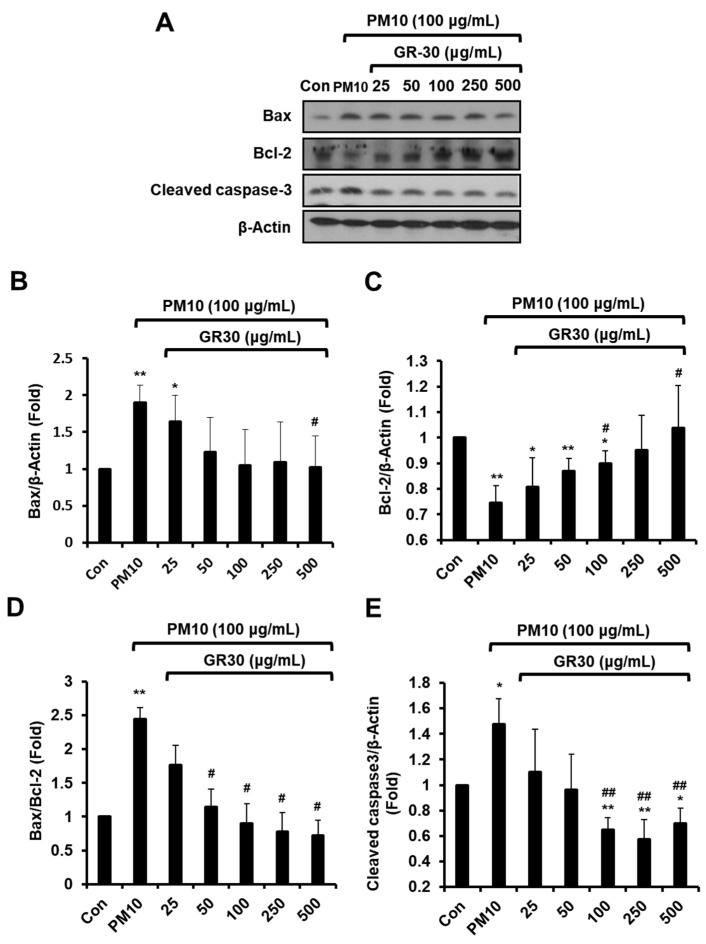
GR30 antagonizes PM10-induced apoptotic cell death via mitochondria-dependent apoptosis in RBL-2H3 cells. (**A**) Expressions of Bax, Bcl-2, and cleaved caspase-3 were examined by Western blot analysis using β-actin as a loading control. Protein levels of (**B**) Bax, (**C**) Bcl-2, and (**E**) cleaved caspase-3 were quantified by densitometric analysis, and the (**D**) Bax/Bcl-2 ratio was determined. The data are expressed as the fold normalized to the untreated control (Con). Each value was presented as the means ± SD (*n* = 3). *, *p* < 0.05; **, *p* < 0.01, compared to the control cells (Con). #, *p* < 0.05; ##, *p* <0.01, compared to the PM10-treated cells (PM10).

**Figure 4 plants-11-02485-f004:**
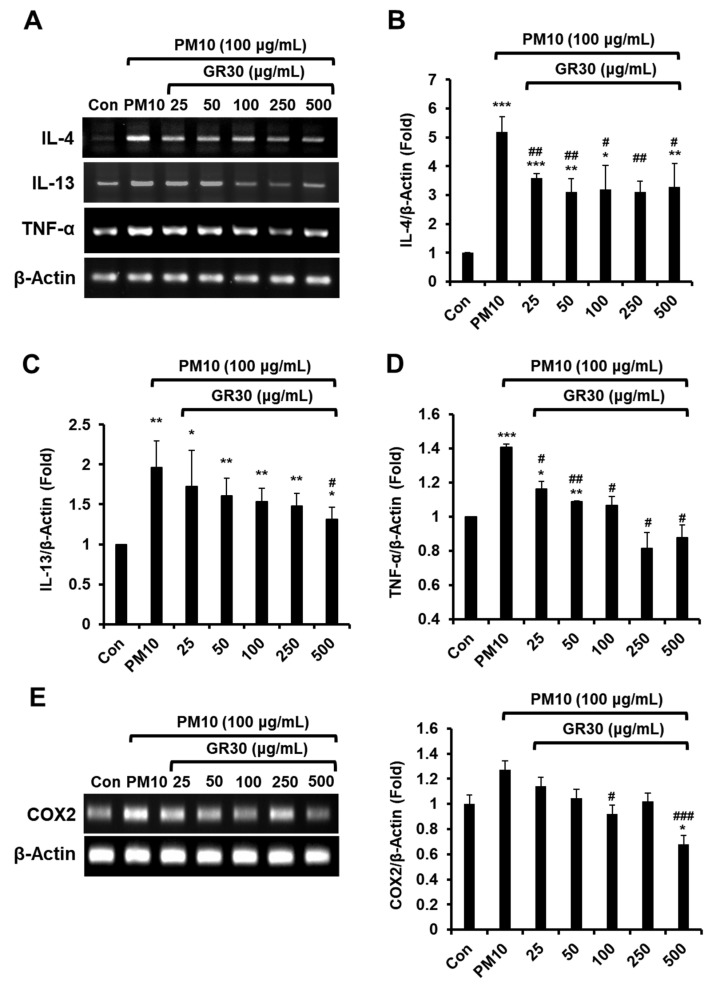
GR30 suppresses PM10-induced IL-4, IL-13, TNF-α cytokine and COX2 expression in RBL-2H3 cells. (**A**) Cells were pretreated with GR30 (25–500 μg/mL) for 2 h and then incubated with PM10 for 48 h. (**B**) IL-4, (**C**) IL-13, (**D**) TNF-α, and (**E**) COX2 mRNA levels were evaluated by RT-PCR and expressed as the ratio of the densitometric measurement of mRNA to the internal standard β-actin. Each value was presented as the means ± SD (*n* = 3). *, *p* < 0.05; **, *p* < 0.01; ***, *p* < 0.001, compared to the control cells (Con). #, *p* < 0.05; ##, *p* < 0.01; ###, *p* < 0.001, compared to the PM10-treated cells (PM10).

**Figure 5 plants-11-02485-f005:**
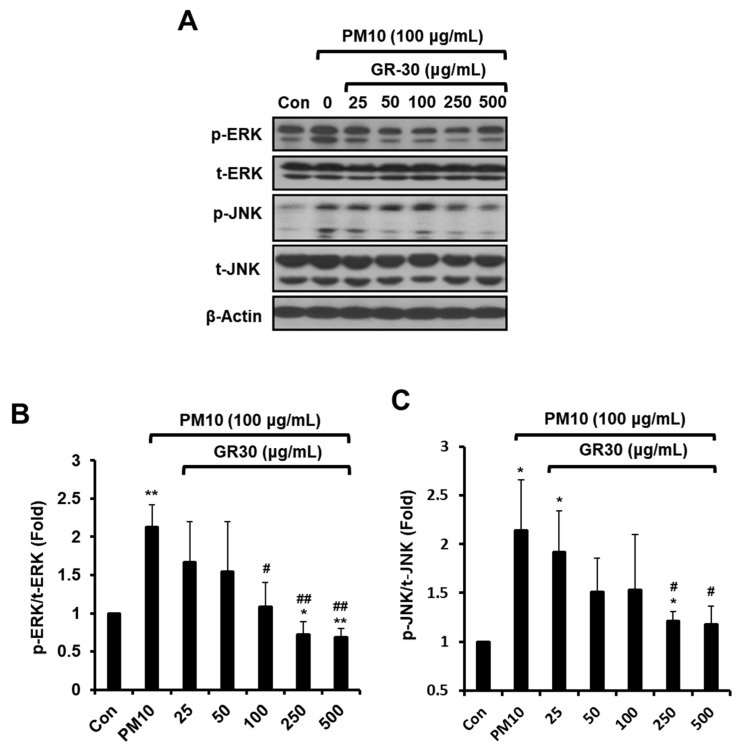
GR30 downregulates the PM10-induced MAPK pathway activation. (**A**) The phosphorylation of MAPK in RBL-2H3 cells was evaluated by Western blot assay. Total ERK (t-ERK) and total p38 (t-JNK) were used as loading controls for phosphorylated ERK (*p*-ERK) and phosphorylated *p*-JNK (*p*-JNK), respectively. The levels of (**B**) *p*-ERK and (**C**) *p*-JNK were quantified by densitometric analysis and expressed as the ratios of the phosphorylated proteins to the corresponding total protein. Each value was presented as the means ± SD (*n* = 3). *, *p* < 0.05; **, *p* < 0.01, compared to the control cells (Con). #, *p* < 0.05; ##, *p* < 0.01, compared to the PM10-treated cells (PM10).

**Figure 6 plants-11-02485-f006:**
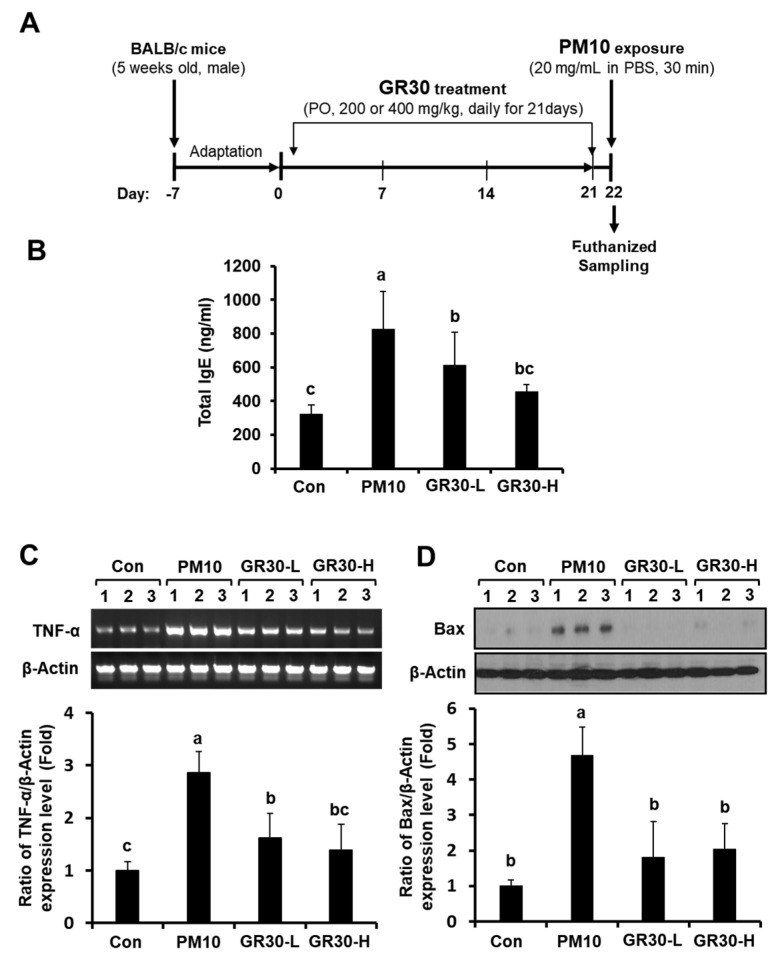
GR30 suppresses PM10-induced serum IgE production and TNF-α and Bax expression in lung tissues. (**A**) Protocol for animal treatment. Mice were orally administered PBS only (control group, PM10 group), 200 mg/kg/day or 400 mg/kg/day GR30 every day for 21 consecutive days. The mice were exposed to PM10 (20 mg/mL, *w*/*v*, in PBS) for 30 min by inhalation using a compressor nebulizer. Effects of GR30 on the levels of (**B**) serum IgE, (**C**) TNF-α mRNA expression and (**D**) Bax protein expression in PM10-challenged mice. The values are expressed as the means ± SD (*n* = 8), and the means with different letters are significantly different from each other (*p* < 0.05), as determined by one-way ANOVA followed by Dunnett’s multiple comparison test. Con, untreated normal control group; PM10, PM10-inhaled control group; GR30-L, GR30 (200 mg/kg)-treated group; GR30-H, GR30 (400 mg/kg)-treated group.

**Figure 7 plants-11-02485-f007:**
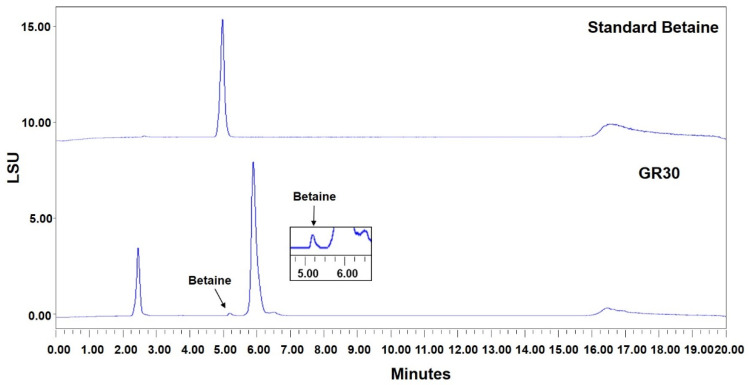
HPLC analysis for betaine in GR30. Betaine in the GR30 preparation was detected and quantified by HPLC system equipped with an ELSD detector. Authentic betaine was used as the reference.

**Figure 8 plants-11-02485-f008:**
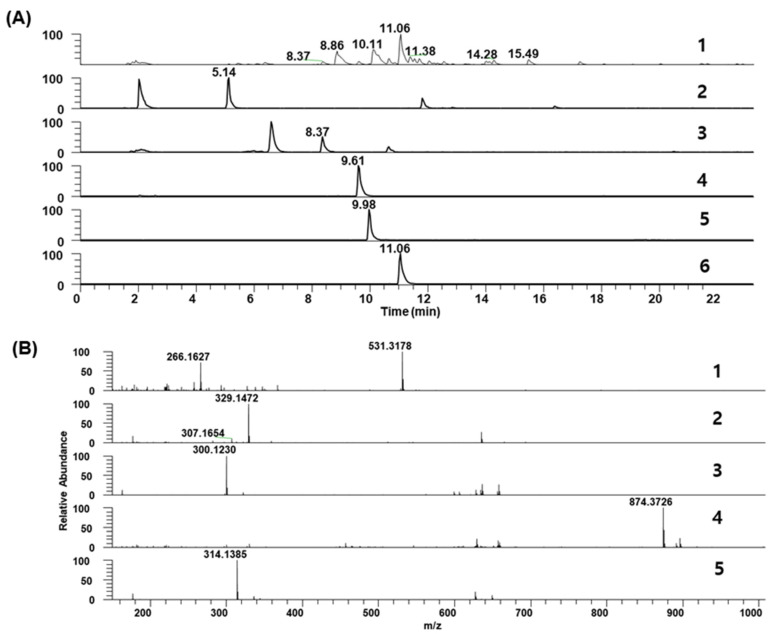
Identification of major anti-inflammatory compounds in GR30 by LC-MS analysis. (**A**) UHPLC chromatographic separation of compounds in GR30 and extracted ion chromatogram (XIC) of 5 compounds (P1~P5), eluted at RT of 5.14 min (P1), 8.37 (P2), 9.61 (P3), 9.98 (P4), and 11.06 (P5), respectively. (**B**) MS spectrum of P1~P5 compounds. Compounds were identified as kukoamine A or kukoamine B (P1), 4-[4-(tert-butoxycarbonyl)piperazin-1-yl]benzoic acid (P2), *n*-caffeoyltyramine (P3), lyciumin A (P4), and coumarin 314 (P5), respectively, based on the high-resolution mass spectrum and MS/MS production ions and summarized in Table 1.

**Table 1 plants-11-02485-t001:** Chemical profile of major compounds in GR30.

Peaks	RT (minute)	m/z ([M + H]^+^)	Formula ([M + H]^+^)	Δppm	Compound
P1	5.14	531.3178	C28 H43 O6 N4	0.11	Kukoamine A or Kukoamine B
P2	8.37	329.1472 ([M + Na]^+^)	C16 H23 O4 N2	0.509	4-[4-(tert-Butoxycarbonyl)piperazin-1-yl]benzoic acid
P3	9.61	300.123	C17 H18 O4 N	0.018	*n*-Caffeoyltyramine
P4	9.98	874.3726	C42 H52 O12 N9	−0.428	Lyciumin a
P5	11.06	314.1385	C18 H20 O4 N	−0.683	Coumarin 314

**Table 2 plants-11-02485-t002:** The primer sequences and cycling parameters used for RT-PCR.

Genes	Primer Sequence ^a^ (F/R 5′ to 3′)	Location ^b^	Size ^c^	Cycling Parameters	Cycles
rat TNF-α	F: CGGAATTCGGCTCCCTCTCATCAGTTC R: GCTCTAGACCCTTGAAGAGAACCTGGG	F: 344 R: 573	230	94 °C 20 s, 60 °C 30 s, 72 °C 30 s	35
rat IL-4	F: ACCTTGCTGTCACCCTGTTC R: TTGTGAGCGTGGACTCATTC	F: 17 R: 307	291	94 °C 20 s, 60 °C 30 s, 72 °C 30 s	35
rat IL-13	F: GCTCTCGCTTGCCTTGGTGGTC R: CATCCGAGGCCTTTTGGTTAGAG	F: 31 R: 304	274	94 °C 20 s, 60 °C 30 s, 72 °C 30 s	35
rat COX2	F: TGACTTTGGCAGGCTGGATT R: ACTGCACTTCTGGTACCGTG	F: 2945 R: 3064	120	94 °C 20 s, 55 °C 30 s, 72 °C 30 s	35
rat β-actin	F: AGCTATGAGCTGCCTGACG R: GGATGCCACAGGATTCCA	F: 793 R: 901	109	94 °C 20 s, 55 °C 30 s, 72 °C 30 s	35
mouse TNF-α	F: GGCAGGTCTACTTTGGAGTCATTGC R: ACATTCGAGGCTCCAGTG AATTCGG	F: 732 R: 1007	276	94 °C 30 s, 55 °C 30 s, 72 °C 40 s	35
mouse β-actin	F: TGCTGTCCCTGTATGCCTCT R: AGGTCTTTACGGATGTCAACG	F: 525 R: 967	443	94 °C 30 s, 55 °C 30 s, 72 °C 40 s	35

^a^ Primer sequences are presented in 5′ to 3′ direction: F, forward; R, reverse. Primer concentration was 1 pM. ^b^ Location of the primer according to the full genomic sequence of rat and mouse: rat TNF-α, NM_012675.3; rat IL-4, AY496861.1; rat IL-13, XM_032911652.1; rat COX2, S6772.1; rat β-actin, NM_031144; mouse TNF-α, BC117057; mouse β-actin, NM_007393. ^c^ Size, PCR product size (base pair).

## Data Availability

Not applicable.

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
