# Peer review of "Anti-Apoptotic and Anti-Inflammatory Effects of an Ethanolic Extract of Lycium chinense Root against Particulate Matter 10-Induced Cell Death and Inflammation in RBL-2H3 Basophil Cells and BALB/c Mice"

_plants, 2022, doi:10.3390/plants11192485_

Round 1

Reviewer 1 Report

The paper titled: Anti-Apoptotic and Anti-Inflammatory Effects of an Ethanolic Extract of Lycium chinense Root against Particulate Matter 10-Induced Basophil Cell Death and Inflammation submitted to plants discussed and confirmed  the potential impacts of Lycium chinense Root  as immuno-modulator for health in vitro and vivo thrugh antiapoptotic and anti-inflammatory pathways.

I have some major points that must be followed for final publishing

1- In introduction: Lines 62-66 What the relation of other types of leukocytes with environmental pollutants?

2- In results: Lines 111-113, Give Scienetific explanation for this contradiction. Do you need additional experiements or What?

3- There are misplacing and miss-writing for figures 2 and 3 recheck???

4- Immunohistochemistry is needed to enforce your results.

5- Section 2.2 and 2.3 must be be adjusted as figures are miss placed.

6-  Level of significance is bot clear and noticeable in some figures such as BAX and BCl2 relative to beta actin. Why?

7- How about TGF-beta1 and COX2 and NFkB Have you examined?

Author Response

Dear Reviewer #1,

A revised version of the manuscript (plants-1883084) entitled “Anti-Apoptotic and Anti-Inflammatory Effects of an Ethanolic Extract of Lycium chinense Root against Particulate Matter 10-Induced Basophil Cell Death and Inflammation” is enclosed for your reconsideration. All the comments from the reviewers have been addressed as suggested and we carefully reviewed our manuscript and tried to do our best to proof read and correct grammars and typos throughout the manuscript. Following editor’s and reviewer’s suggestions, several additional experiments were performed and several sentences were added. Accordingly, the manuscript was substantially modified and improved. Please note that, following reviewer’s suggestion, the title of the manuscript was changed to “Anti-Apoptotic and Anti-Inflammatory Effects of an Ethanolic Extract of Lycium chinense Root against Particulate Matter 10-Induced Cell Death and Inflammation in RBL-2H3 Basophil cells and BALB/c mice” in the revised version. We appreciate Editor and reviewers very much for valuable comments and suggestions to strengthen our manuscript. All changes in the revised manuscript are highlighted in red. (Please find the details of our answer from the uploaded word file).   

Reviewer 2 Report

The manuscript is focused on the investigation of anti-apoptotic and anti-inflammatory effects of an extract from Lycium chinense roots, using both in vitro and in vivo methods. The study is complex and the results are interesting, suggesting protective effects of natural compounds from Lycium against PM-induced toxicities and adding new data concerning possible mechanisms of action. Only a few minor issues need to be corrected/better explained in the manuscript:

1. In the in vivo test, from Figure 6A it is presumed that animals were sacrificed at the end of week 4  but in section 4.4, the authors state that "on day 22 the animals were euthanized". Please clarify this issue.

2. Also in section 4.4, please use common international names for the substances used to euthanize the animals and not commercial names like Zoletil or Rompun.

Author Response

Dear Reviewer #2,

A revised version of the manuscript (plants-1883084) entitled “Anti-Apoptotic and Anti-Inflammatory Effects of an Ethanolic Extract of Lycium chinense Root against Particulate Matter 10-Induced Basophil Cell Death and Inflammation” is enclosed for your reconsideration. All the comments from the reviewers have been addressed as suggested and we carefully reviewed our manuscript and tried to do our best to proof read and correct grammars and typos throughout the manuscript. Following editor’s and reviewer’s suggestions, several additional experiments were performed and several sentences were added. Accordingly, the manuscript was substantially modified and improved. Please note that, following reviewer’s suggestion, the title of the manuscript was changed to “Anti-Apoptotic and Anti-Inflammatory Effects of an Ethanolic Extract of Lycium chinense Root against Particulate Matter 10-Induced Cell Death and Inflammation in RBL-2H3 Basophil cells and BALB/c mice” in the revised version. We appreciate Editor and reviewers very much for valuable comments and suggestions to strengthen our manuscript. All changes in the revised manuscript are highlighted in red. (Please find the details of our answers from the uploaded word file.)

Author Response

Dear Reviewer #3,

A revised version of the manuscript (plants-1883084) entitled “Anti-Apoptotic and Anti-Inflammatory Effects of an Ethanolic Extract of Lycium chinense Root against Particulate Matter 10-Induced Basophil Cell Death and Inflammation” is enclosed for your reconsideration. All the comments from the reviewers have been addressed as suggested and we carefully reviewed our manuscript and tried to do our best to proof read and correct grammars and typos throughout the manuscript. Following editor’s and reviewer’s suggestions, several additional experiments were performed and several sentences were added. Accordingly, the manuscript was substantially modified and improved. Please note that, following reviewer’s suggestion, the title of the manuscript was changed to “Anti-Apoptotic and Anti-Inflammatory Effects of an Ethanolic Extract of Lycium chinense Root against Particulate Matter 10-Induced Cell Death and Inflammation in RBL-2H3 Basophil cells and BALB/c mice” in the revised version. We appreciate Editor and reviewers very much for valuable comments and suggestions to strengthen our manuscript. All changes in the revised manuscript are highlighted in red. (Please find the details of our answers from the uploaded word file.)

Round 2

Reviewer 1 Report

The paper has been corrected and can be pass for publication in plants but still need minor modifications. Data for COX2 must be included to complete the story and to be sound.

Some minor linguistic points need your attention.

Author Response

REVIEWERS' COMMENTS:

Reviewer #1:

The paper has been corrected and can be pass for publication in plants but still need minor modifications. Data for COX2 must be included to complete the story and to be sound.

Response to comments of Reviewer #1:

-> Ans) Following reviewer’s suggestion, we have decided to include the results on the effects of PM10 and GR30 on the COX2 mRNA expression to the revised manuscript. Accordingly, several sentences on COX2 were added or rephrased in the sections of Methods & materials, Results, and Discussions in the revised manuscripts. Figure4 was also redrawn and a couple of references related to COX2 was also added to the Reference list. Please find details on COX2 story from the uploaded file. Again, we appreciate very much the Reviewer #1 for valuable comments and suggestions to strengthen our manuscript.
